# Deep Learning and Its Applications in Computational Pathology

**Runyu Hong** *[ID] and David Fenyö *[ID]

Institute for Systems Genetics, NYU Grossman School of Medicine, New York, NY 10016, USA
* Correspondence: Runyu.Hong@nyu.edu (R.H.); david@fenyolab.org (D.F.)

**Abstract:** Deep learning techniques, such as convolutional neural networks (CNNs), generative adversarial networks (GANs), and graph neural networks (GNNs) have, over the past decade, changed the accuracy of prediction in many diverse fields. In recent years, the application of deep learning techniques in computer vision tasks in pathology has demonstrated extraordinary potential in assisting clinicians, automating diagnoses, and reducing costs for patients. Formerly unknown pathological evidence, such as morphological features related to specific biomarkers, copy number variations, and other molecular features, could also be captured by deep learning models. In this paper, we review popular deep learning methods and some recent publications about their applications in pathology.

**Keywords:** deep learning; machine learning; histopathology; computational pathology; convolutional neural networks; generative adversarial networks

## 1. Introduction

With the development of artificial intelligence and machine learning techniques in the past decade, many deep-learning-based computer vision models are playing important roles in daily life, and have revolutionized various industries through their superior performance and efficiency in prediction tasks, such as autopilot, machine translation, electronic sports, and biometry [1–6]. Recently, these technologies have also shown their extraordinary potential and capabilities in solving many complicated questions in the biomedical field by analyzing massive amounts of biomedical data, such as protein structural predictions with Alphafold which outperforms experimental results [7,8], and tumor segmentation in MRI scans [1]. In particular, computational pathology, a discipline that involves the effort of both pathologists and informaticians, has especially benefitted from the advancement of deep learning in recent years [9–12]. Several models have also been demonstrated to be useful in clinical diagnoses based on histopathology images [13–15]. In addition, some model-extracted morphological features show correlations with features at a molecular level, including single mutations and subtypes, most of which are previously unknown to human pathologists and clinicians [14]. Here, we discuss these deep learning methods from a technical perspective and summarize their successful applications in pathology from recent publications.

## 2. Deep Learning Techniques

### 2.1. Convolutional Neural Networks

Deep learning is a type of machine learning method that is using a multi-layer perceptron called artificial neural networks (ANN) [1,2,16]. Training a deep learning model involves designing and selecting a neural network architecture, loss functions, and evaluation metrics, as well as tuning the hyperparameters of batch size, step size, and regularization methods [1,2,17,18]. Convolutional neural networks (CNNs), variants of ANNs, have proved their power in tackling various computer vision tasks, such as image classification, segmentation, and object detection [19–23]. The first modern CNN architecture, LeNet5, was introduced by Yann LeCun et al. in 1998 [24]. This gradient-based six-layer

convolutional neural network shows its power in recognizing hand-written digits and characters [24]. However, the development of CNNs was restricted by limited computational compacities and resources for over a decade. The advancement of computational hardware in recent years, especially graphical processing units (GPUs) and tensor processing units (TPUs), empowers the development of deep neural networks. Many CNN architectures, such as AlexNet [25], VGG [26], InceptionNet [20], and ResNet [27], can be trained into models that even outperform human beings in a computer vision classification challenge called ImageNet, which contains 1.2 million high-resolution images of more than 1000 classes [19,27–30].

AlexNet was introduced in 2012. This architecture is much larger than the previous LeNet5, with 650,000 neurons and 60 million trainable parameters packed into this design, with 5 convolutional and 3 fully connected layers [25]. Overlapping max pooling, ReLU nonlinearity, and dropout regularization are also incorporated. Due to the development of hardware, AlexNet at that time could only be trained and run on two GPUs [25]. It achieved a top-five test error rate of 15.3%, which made it the winner of the ILSVRC-2012 competition and outperformed the second-best model by more than 10% [25]. The overwhelming success of AlexNet drew people's attention back to CNNs, and numerous other new architectures, including the VGG, InceptionNet and ResNet, were developed in the following years.

VGG architecture was introduced in 2014, when it won the ImageNet challenge [26]. Compared with AlexNet, VGG increases the depth of the model by adding more convolutional layers with smaller convolutional filters [26]. However, with the introduction of newer architectures in the following years, VGG architecture has lost its popularity due to the gigantic size, high complexity to train, and less accurate performance.

InceptionV1 architecture was announced in 2015, with the name GoogLeNet, which is a 22-layer deep CNN (Figure 1) [20]. The two key innovations that make Inception architectures outstanding are the inception module and auxiliary classifier. The inception module consists of multiple convolutional kernels with different sizes on the same layer [20]. This design allows the model to capture similar features of various sizes. Deep CNNs are prone to overfitting and passing gradient updates through the entire network is hard, which is often referred to as the vanishing gradient problem. By adding auxiliary classifiers in the middle of the network, the auxiliary loss from the middle of the model is taken in the final loss calculation, so that the gradients also represent the middle part of the network [20]. InceptionV2 and InceptionV3 were introduced in 2016, which modified the inception module by factorizing the larger kernels into a stack of smaller kernels to make the architecture more computationally efficient [28]. In addition, InceptionV3 uses the RMSProp optimizer and adds batch normalization into the auxiliary classifiers, significantly improving the performance, with a top-five error of 3.57% and top-one error of 17.2% on ImageNet, much better than a human [28]. InceptionV4 further refined the architecture by adding reduction blocks and unifying the inception modules [29].

A major competitor of InceptionNet is Resnet, which applies the idea of residual connection (Figure 2) [27]. In this architecture, each layer learns the residuals from the previous layer with reference to the layer inputs [27]. The top-five error on ImageNet is 3.57%, which is similar to the performance of InceptionV3 [27]. Interestingly, this residual connection idea was later adapted by the InceptionNet team to develop InceptionResNetV1, a modified version of InceptionV3, and InceptionResNetV2, a modified version of InceptionV4 [29]. Using the residual connection, InceptionResNetV2 achieved a markedly improved 3.1% top-five error on ImageNet [29].

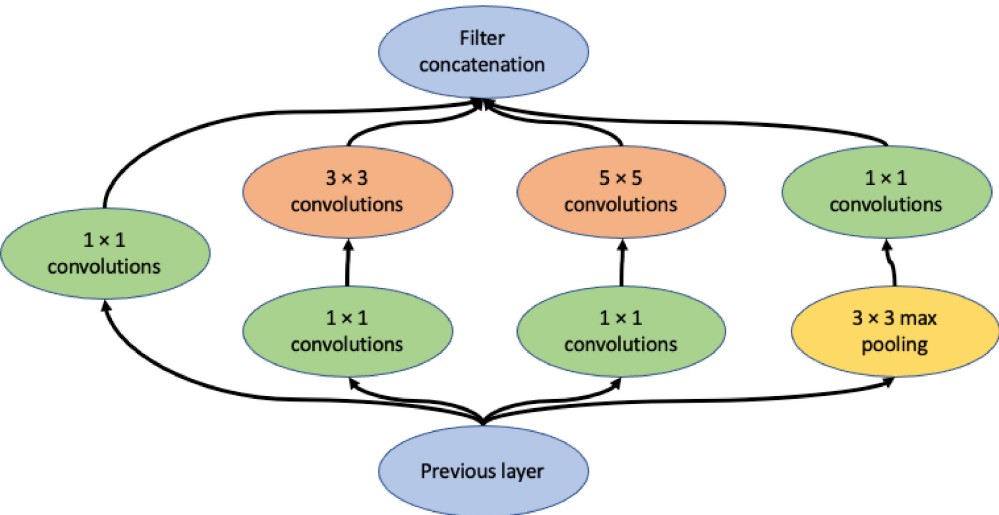

**Figure 1.** Diagram of the inception module [19], containing 4 branches with convolutional kernels of different sizes.

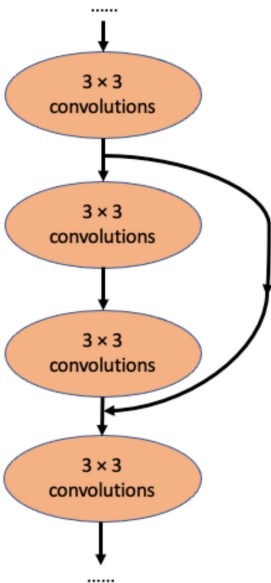

**Figure 2.** The concept of residual connection in ResNet [27]. Here, the middle 2 convolutional layers are skipped by the residual connection.

### *2.2. Visualization of CNN Models*

Ever since the introduction of deep neural networks, people have been eager to know what their models have learned [31]. For image-based tasks with CNN models, visualizing the captured features is the most straightforward way. Class activation mapping (CAM) and saliency maps are two simple ways to visualize the learned features by projecting the weights and gradients of the output layer back to the input image [32–34]. However, these visualization methods are image-specific and will only roughly imply where the models are focusing. In addition, many saliency methods have been criticized recently for giving misleading visualization interpretations, and researchers are advised to use them with caution [35]. To unveil the CNN models further, direct deconvolution and indirect optimization are the two major approaches [36]. Deconvolution starts with finding an image from the dataset that triggers high activity to the neuron of interest and the gradient of neuron activity is calculated [36]. In general, a deconvolutional network is a reversed convolutional network, which maps features back to pixels [37]. However, deconvolution

visualization can be noisy and may contain features that are not easy to interpret [38]. The indirect optimization approach can provide more accurate visualization than that from deconvolution [39]. The algorithms optimize the colors of the pixels of an image to maximize activation of the neuron of interest [38–40]. Once a set of optimized images for many neurons has been obtained, a dimensional reduction visualization method, such as UMAP and tSNE, can create an atlas that systematically displays the correlations of features captured by different neurons at the same layer [38,41–43].

*2.3. Graph Neural Networks*

Graph neural networks (GNNs) are a type of neural network which deal with data consisting of relational information [44]. Data with a non-Euclidean structure of information, such as particle interactions, molecular structures, and object relationships in images, could be modeled by GNNs [45]. In general, GNNs can be further classified into four categories: recurrent GNNs, convolutional GNNs, graph autoencoders, and spatial–temporal GNNs [45].

*2.4. Generative Adversarial Networks*

Generative adversarial networks (GANs) are a type of neural network consisting of two networks that are trained at the same time [46]. The generator part is trained to create fake images which tries to fool the discriminator, while the discriminator, trained with both real and generated fake images, is able to distinguish them [46]. Many variants of GANs have been applied to different tasks, such as style transfer, the visualization of neural networks, and object segmentation [46–53]. Cycle-GAN, a GAN variant using cycle-consistence loss to train two pairs of generators and discriminators simultaneously, has become increasingly popular for image-to-image translation tasks [50]. Unlike conditional GANs, which require two styles of paired data, cycle-GANs only need two sets of images of two styles, which significantly lowers the data requirements while preserving the quality of style transfer [50].

## 3. Applications in Computational Pathology

*3.1. Classification and Feature Prediction*

With the success of CNN models in various real-world computer-vision classification tasks, researchers and scientists have also trained and tested these models in case scenarios in biomedical fields, including pathology. These studies may involve training an existing CNN architecture from scratch. However, it requires more data, and the data augmentation techniques may not always be suitable for biomedical images. Alternatively, transfer learning techniques, which freeze most of the parameters from a model often pre-trained on ImageNet, have more advantages in terms of the data size requirements. For example, an InceptionV3-based ImageNet pre-trained CNN model can achieve a high level of accuracy in determining skin lesion malignancy and the possibility of melanoma [13,54].

In clinical settings, pathologists typically examine histopathology slides under microscopes to provide diagnosis or other clinical information. Due to the development of digital pathology equipment, digitizing histopathology slides is cheaper and more accessible. As a result, more and more deidentified digital histopathology slide images have become available in many databases. These images, often with extremely large dimensions, are saved in special image file formats (e.g., .svs or .scn), which is a tuple of the same image with different resolutions [55]. Thus, in order to fit these digital histopathology images into CNN architectures, people usually develop their own customized pipelines with commonly used techniques, such as tiling the whole slide images (WSI) or sampling regions of interest (ROIs) (Figure 3) [56]. In the past few years, classification CNN models trained on histopathology images have shown phenomenally high performance and promising clinical potential in predicting both morphological features and molecular features. The visualization techniques also reveal results that often match pathologists' expectations and many models are generalizable to independent real-world clinical images. For example,

Inception and InceptionResNet architectural models demonstrate high accuracy and other statistical metrics in predicting subtypes and key biomarker mutations, such as STK11 and EGFR, in non-small-cell lung cancer histopathology slides [14,57,58]. With the integration of other critical clinical variables and images, immune response, G-CIMP, and telomere length can be predicted in glioblastoma patients [59]. BRAF mutation, a well-known biomarker in malignant melanoma, can also be accurately predicted with a CNN-based model [60]. Other molecular and genomic features, such as microsatellite instability (MSI), can be predicted from histopathology slides with a reasonable accuracy as well [15]. The critical gene expression level could also be inferred by applying these CNN classification models to WSI [61]. Some contemporary models also show successful classification results in the histopathology images of multiple tissue types [62,63]. These successful cases indicate that CNNs represent a suitable approach to study the correlation between molecular features and morphological features in histopathology slides, some of which may be undetectable or often ignored by human pathologists.

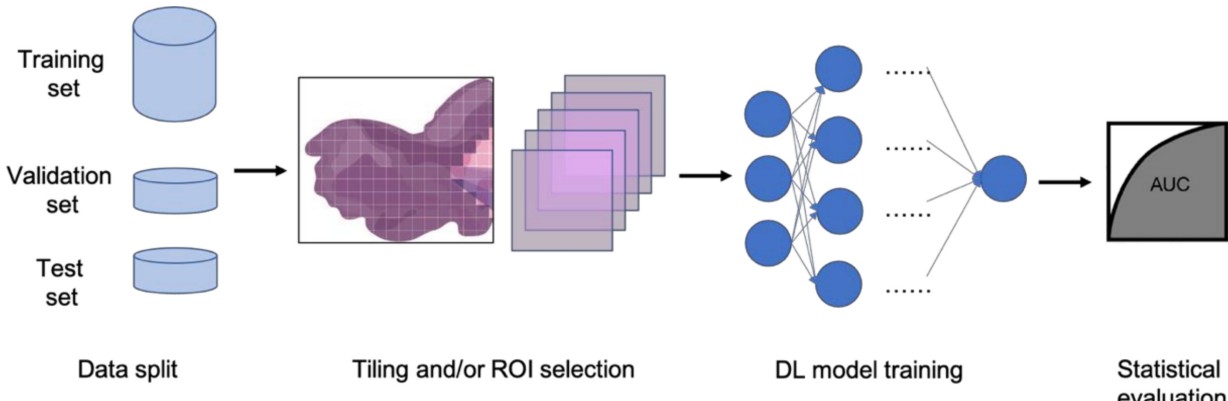

**Figure 3.** Typical classification model pipeline for histopathology images.

However, histopathology images are quite different from the images in the ImageNet because of their extremely large sizes, higher resolution, and sparser useful feature distributions [11,55,64]. Deep learning architectures that could take advantage of these characteristics are very likely to achieve better results, unveiling more interesting hidden features in histopathology image classification tasks. For instance, a multi-resolution CNN model, which takes advantage of the data structure of .svs and .scn image files, achieves higher performance in classifying endometrial cancer molecular features than its single resolution counterparts [65]. Weakly supervised techniques, such as multiple instance learning, also demonstrate decent performance in classification tasks of histopathology images, and have gained popularity in recent years [64,66,67]. The innovative idea of bringing GNN models into solving histopathology classification problems develops greater capacity in understanding the subtle relationships between features of different tissue structures and at different locations on giant digital histopathology slides [68,69].

### 3.2. Segmentation

In addition to classification tasks, CNN models are also capable of segmenting cells or tissue in histopathology slides [9,55]. The segmented cells or tissue could then be used to train classification models for different prediction tasks, including the recurrence of non-small-cell lung cancer [70] and endometrial tissue types [71]. A popular segmentation CNN architecture used in the biomedical field is U-net, which has a similar structure to an autoencoder [72]. A 3D version of U-net, which has 3D convolutional layers instead of 2D convolutional layers, is capable of segmenting volumetric images [22]. Modified U-net architectures, such as USE-net [73], Focus U-net [74], and U-net with attention gate [75], have achieved even better performance in various biomedical image segmentation tasks than vanilla U-net. Other autoencoder-based methods have also achieved promising

results in segmentation tasks of histopathology images, such as highlighting tumor regions in liver cancer WSI [76]. Well-trained style transfer models are also viable options for segmentation tasks [48]. With the introduction of GAN, using conditional-GAN or cycle-GAN models and in combination with CNN models for segmentation problems is also shown to be viable, with less stringent training data requirements [46,53,77]. Unlike most classification models, the segmentation models can be more adaptive to different types of tissues due to the similarities of the stained features and textures of the histopathology slides [78]. Additionally, the evaluation metrics of these classification models can be drastically different from those of the classification models. The segmentation labels are also usually images; therefore, it is not easy to determine a binary prediction or even a prediction score at the per-image level. Hence, typical statistical metrics, such as AUROC or precision and recall, are often not capable of fairly evaluating segmentation tasks. Pixel-level metrics, such as intersection over the union (IoU), also pose weaknesses because it cannot objectively give relative importance to pixels of different regions. Object-level metrics can be an optimal alternative, but the requirement of identifying all objects on the label images prohibits its adoption in real-world model evaluation. Therefore, researchers often use customized evaluation metrics with a combination of customized pixel weights, dice loss, and IoU with specific thresholds [79,80].

## 4. Summary

In this review paper, we have introduced popular deep learning algorithms, CNN, GNN, and GAN, and also highlighted mechanisms of how they work and how they can be applied to solve clinical and scientific questions in pathology. We have also discussed recent publications which show that these deep learning techniques have the potential to be useful in classifying or segmenting histopathology imaging data. With the continuing advancement of machine learning and deep learning techniques and the development of hardware and software, it is realistic to believe that the integration of artificial intelligence and pathology will become an even more attractive field to explore. Compared with conventional computational methods, deep learning techniques generally run faster and have much better performance in pathology tasks. Although one has to be rigorous and ethical about translating these AI-based technologies into clinical settings, we still hold an optimistic view that they will eventually revolutionize medical diagnosis processes and really push the development of precision medicine forward. Above all, the ultimate goal of introducing AI into pathology and biomedicine in general is to make healthcare more accessible, affordable, and agreeable.

Nevertheless, there are still a number of limitations of the current studies and potential obstacles which prevent these models from implementation in contemporary real-world clinical settings. For example, only patterns with prior understanding from pathologists can be used as reliable evidence for prediction, which significantly limits the tasks for which deep learning models can be applied. In addition, the patient samples that can be used as training, validation, and test sets are also very limited for each of the specific tasks of interests. Moreover, detailed labels of medical images are often not available, and the labeling standards among clinicians also vary significantly in different countries. Additionally, the interpretability of deep learning models applied to histopathology images remains debatable, especially among clinicians. More advanced self-supervised or semi-supervised methods may solve some of these problems from a technical perspective in the future.

**Author Contributions:** Conceptualization, R.H. and D.F.; methodology, R.H. and D.F.; resources, R.H.; data curation, R.H.; writing—original draft preparation, R.H.; writing—review and editing, D.F.; visualization, R.H.; supervision, D.F.; project administration, D.F.; funding acquisition, D.F. All authors have read and agreed to the published version of the manuscript.

**Funding:** This work was supported by NIH/NCI U24CA210972.

**Institutional Review Board Statement:** Not applicable.

**Informed Consent Statement:** Not applicable.

**Data Availability Statement:** Not applicable.

**Acknowledgments:** We would like to thank all members of the Fenyö laboratory and the administration team of ISG at NYU.

**Conflicts of Interest:** The authors declare no conflict of interest.

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
