# Peer review of "Deep Learning and Its Applications in Computational Pathology"

_biomedinformatics, doi:10.3390/biomedinformatics2010010_

Round 1

Reviewer 1 Report

Dear Authors, you did a good job summarizing the algorithms and applications of artificial neural networks in pathology. However, there are a few points where I think some revision will contribute to the comprehensiveness of the paper, as follows.

  1. Figure 2 needs to be better formatted and annotated, for example, get the plot centered, and annotate the different layers with caption of how different layers of residuals are connected.
  2. The application of artificial neural networks in genomics and disease prediction is a trending topic. Are you constricting your concept of pathology in histopathology slides or a generalized concept including other areas such as genomics and functional magnetic resonance imaging?
  3. The interpretability of artificial neural networks, especially in biology and pathology, has remained debatable. Some discussion upon this is needed for audience to better understand the application.
  4. How do artificial neural networks compare with traditional methods and human in terms of performance and speed? A little discussion is needed on this too.

Hopefully these points will add to the comprehensiveness of the paper.

Author Response

We would like to thank reviewer for their comments. Here are our point-to-point response.

  1. Figure 2 needs to be better formatted and annotated, for example, get the plot centered, and annotate the different layers with caption of how different layers of residuals are connected.

We have centered Fig 2 and added description in the caption. Technically, the residual connection are the same across the architecture, so we only put an example of residual connection.

  1. The application of artificial neural networks in genomics and disease prediction is a trending topic. Are you constricting your concept of pathology in histopathology slides or a generalized concept including other areas such as genomics and functional magnetic resonance imaging?

The topic of this review paper is specifically focusing on NN applications in computational histopathology imaging. While genomics and MRI are both interesting topics, from our point of view, they may not be good fits for this paper.

  1. The interpretability of artificial neural networks, especially in biology and pathology, has remained debatable. Some discussion upon this is needed for audience to better understand the application.

The last paragraph of the paper was mainly talking about the limitations of NN techniques nowadays in pathology. We also added a sentence to highlight the controversy of the interpretability.

  1. How do artificial neural networks compare with traditional methods and human in terms of performance and speed? A little discussion is needed on this too.

We added the discussion comparing deep learning techniques and the conventional computational methods in the summary section.

Reviewer 2 Report

An interesting review focusing on deep-learning architectures applied to different clinical scenarios. 
In particular, it is a good work, providing a fairly comprehensive view of architecture (CNN, GNN, GAN) applied to different pathological contexts.

In order to further improve this contribution there are some comments to be approached appropriately.

---------------------------------------------

Comment 1 (section  '1. Introduction')
To provide a more general introductory scenario, I suggest to rephrase the sentence "... have revolutionized various industries by their superior performance and efficiency in prediction tasks, such as autopilot, machine translation, electronic sports, and facial recognition " as follow: "... have revolutionized various industries by their superior performance and efficiency in prediction tasks, such as autopilot, machine translation, electronic sports, and biometry". Contextually, I suggest to add the following literature work
  - Militello, C. et al. (2021). Fingerprint Classification based on Deep Learning Approaches: Experimental Findings and Comparisons. (2021). Symmetry 2021, 13(5), 750; https://doi.org/10.3390/sym13050750

Comment 2 (section '2.1. Convolutional Neural Networks')
Please, add references in the following sentence as indicated: "Many CNN architectures, such as AlexNet [REF], VGG [REF], InceptionNet [REF], and ResNet [REF], can be trained into models that even outperform human beings in a computer vision classification challenge called ImageNet, which contains 1.2 million high-resolution images of more than 1000 classes".

Comment 3 (section '2.1. Convolutional Neural Networks')
By means of figure 1 it is very difficult to understand the architcture of InceptionNet. I suggest to replace this figure with an architectural scheme.
Same problem with figure 2.

Comment 4 (section '2.1. Convolutional Neural Networks')
As well as the InceptionNet and ResNet architectures (or at least part of them) are reported, for the sake of completeness I expect that those of the other CNNs described in the section (i.e. AlexNet and VGG) are also illustrated.

Comment 5 (section 2.3. Graph Neural Networks)
This section is minimal and does not provide an exhaustive picture of this type of neural network. The two referenced papers (i.e. [43] and [44]) are dellereview. I expect the authorizations to focus on specific types and applications of GNN. I suggest extending a little discussion by adding and discussing some other literature work.

Comment 6 (section '3.1. Classification and feature prediction')
"These images, often with extremely large dimension of hundreds of thousands by hundreds of thousands of pixels" -> "These images, often with extremely large dimension".

Comment 7 (section '3.1. Classification and feature prediction')
Please, specify 'SVS' and 'SCN' acronyms.
Also check the entire manuscript to avoid using undefined acronyms. 

Comment 8 (section '3.2. Segmentation')
Regarding the latest advances concerning segmentation, please introduce channel and spatial attention mechanisms by discussing the following highly relevant articles on outstanding U-Net versions:
 - Rundo, L. et al. (2019). USE-Net: Incorporating Squeeze-and-Excitation blocks into U-Net for prostate zonal segmentation of multi-institutional MRI datasets. Neurocomputing, 365, 31-43. DOI: 10.1016/j.neucom.2019.07.006
 - Schlemper, J. et al. (2019). Attention gated networks: learning to leverage salient regions in medical images, Med. Image Anal. 53 (2019) 197–207, DOI: 10.1016/j.media.2019.01.012
 - Yeung, M. et al. (2021). Focus U-Net: A novel dual attention-gated CNN for polyp segmentation during colonoscopy. Computers in Biology and Medicine, 104815. DOI: 10.1016/j.compbiomed.2021.104815

Comment 9 (section '4. Summary')
Authors state "We also discussed recent publications that apply these deep learning techniques to classify or segment histopathology imaging data".
This statement is limiting for the work, which actually discusses and analyzes different pathological contexts that can be found in clinical practice, and that find in deep-learning techniques a useful tool to approach the problems of classification and segmentation of biomedical images. Please rework the sentence appropriately.

Author Response

We would like to thank reviewer for their comments. Here are our point-to-point response.

Comment 1 (section  '1. Introduction')
To provide a more general introductory scenario, I suggest to rephrase the sentence "... have revolutionized various industries by their superior performance and efficiency in prediction tasks, such as autopilot, machine translation, electronic sports, and facial recognition " as follow: "... have revolutionized various industries by their superior performance and efficiency in prediction tasks, such as autopilot, machine translation, electronic sports, and biometry". Contextually, I suggest to add the following literature work
  - Militello, C. et al. (2021). Fingerprint Classification based on Deep Learning Approaches: Experimental Findings and Comparisons. (2021). Symmetry 2021, 13(5), 750; https://doi.org/10.3390/sym13050750

We have modified the introduction as the reviewer suggested.

Comment 2 (section '2.1. Convolutional Neural Networks')
Please, add references in the following sentence as indicated: "Many CNN architectures, such as AlexNet [REF], VGG [REF], InceptionNet [REF], and ResNet [REF], can be trained into models that even outperform human beings in a computer vision classification challenge called ImageNet, which contains 1.2 million high-resolution images of more than 1000 classes".

We have modified as the reviewer suggested.

Comment 3 (section '2.1. Convolutional Neural Networks')
By means of figure 1 it is very difficult to understand the architcture of InceptionNet. I suggest to replace this figure with an architectural scheme.
Same problem with figure 2.

We appreciate the reviewer for suggesting showing the full architectures of Inception and Resnet. However, we would like to point out that the purpose of these 2 figures is to show the inception module and the residual connection, not InceptionNet and Resnet. These building structures are used in other deep learning architectures as well, such as the famous InceptionResnet introduced in 2017. Besides, if the readers are interested in exploring these architectures in detail, they could easily find them in the cited articles of the figure legends. We hope this explanation could address reviewer’s concerns.

Comment 4 (section '2.1. Convolutional Neural Networks')
As well as the InceptionNet and ResNet architectures (or at least part of them) are reported, for the sake of completeness I expect that those of the other CNNs described in the section (i.e. AlexNet and VGG) are also illustrated.

Similar to the response to the previous comments, readers could easily find the full architectures in the cited articles. Here, we just want to highlight the 2 key building structures, inception module and residual connection. AlexNet and VGG are both stacking of convolutional layers, which makes it very hard to show only their key building structures.

Comment 5 (section 2.3. Graph Neural Networks)
This section is minimal and does not provide an exhaustive picture of this type of neural network. The two referenced papers (i.e. [43] and [44]) are dellereview. I expect the authorizations to focus on specific types and applications of GNN. I suggest extending a little discussion by adding and discussing some other literature work.

We appreciate the reviewer’s comment about GNN. The idea of using GNN in computational pathology is quite new, and there are limited numbers of mature publications in that field. Our goal here is to briefly introduce GNN and a possible future research direction of applying it to computational pathology. We could remove this section if the reviewer thinks it is reluctant.

Comment 6 (section '3.1. Classification and feature prediction')
"These images, often with extremely large dimension of hundreds of thousands by hundreds of thousands of pixels" -> "These images, often with extremely large dimension".

We have modified as the reviewer suggested.

Comment 7 (section '3.1. Classification and feature prediction')
Please, specify 'SVS' and 'SCN' acronyms.
Also check the entire manuscript to avoid using undefined acronyms. 

We have added clarification about the file format.

Comment 8 (section '3.2. Segmentation')
Regarding the latest advances concerning segmentation, please introduce channel and spatial attention mechanisms by discussing the following highly relevant articles on outstanding U-Net versions:
 - Rundo, L. et al. (2019). USE-Net: Incorporating Squeeze-and-Excitation blocks into U-Net for prostate zonal segmentation of multi-institutional MRI datasets. Neurocomputing, 365, 31-43. DOI: 10.1016/j.neucom.2019.07.006
 - Schlemper, J. et al. (2019). Attention gated networks: learning to leverage salient regions in medical images, Med. Image Anal. 53 (2019) 197–207, DOI: 10.1016/j.media.2019.01.012
 - Yeung, M. et al. (2021). Focus U-Net: A novel dual attention-gated CNN for polyp segmentation during colonoscopy. Computers in Biology and Medicine, 104815. DOI: 10.1016/j.compbiomed.2021.104815

We have added the reviews for the above U-net based methods.

Comment 9 (section '4. Summary')
Authors state "We also discussed recent publications that apply these deep learning techniques to classify or segment histopathology imaging data".
This statement is limiting for the work, which actually discusses and analyzes different pathological contexts that can be found in clinical practice, and that find in deep-learning techniques a useful tool to approach the problems of classification and segmentation of biomedical images. Please rework the sentence appropriately.

We lowered the tone to be “We also discussed recent publications that show these deep learning techniques have the potential to be useful in classifying or segmenting histopathology imaging data.”

Round 2

Reviewer 1 Report

Dear Authors, thank you for your revision. You have responsively answered my questions. I think the manuscript is now well structured. There are a few places that could be revised for grammatical reasons, listed as follows.

  1. The name of figure 1: "Inception Modules".
  2. Line 168-169: dimension"s", format"s".
  3. It will be better to elaborate the messages in figure captions (both figure 1 & 2).

Hope these will shape the manuscript for better reading experience.

Author Response

We would like to thank the reviewer for their comments. We modified the draft and corrected the minor issues mentioned by the reviewer. 

Reviewer 2 Report

Thanks to the authors for the improvement work done. All concerns raisen by reviewers in the previous review round have been properly approached. The contribution has been corrected/integrated according to the reveived comments.

Now the quality of the work is good for the publication.

Author Response

We would like to thank the reviewer for their recommendation for publication!